# Development of Dual-Targeted Mixed Micelles Loaded with Celastrol and Evaluation on Triple-Negative Breast Cancer Therapy

**DOI:** 10.3390/pharmaceutics16091174

**Published:** 2024-09-06

**Authors:** Siying Huang, Simeng Xiao, Xuehao Li, Ranran Tao, Zhangwei Yang, Ziwei Gao, Junjie Hu, Yan Meng, Guohua Zheng, Xinyan Chen

**Affiliations:** 1Pharmacy Faculty, Hubei University of Chinese Medicine, Wuhan 430065, China; hsy0701sc@163.com (S.H.); xiaosm0430@163.com (S.X.); lxh1114094569@icloud.com (X.L.); tr1132330912@163.com (R.T.); zwyang2018@163.com (Z.Y.); ziweigao1999@163.com (Z.G.); hero0712@163.com (J.H.); yanmeng2016@126.com (Y.M.); 2Hubei Shizhen Laboratory, Wuhan 430065, China

**Keywords:** TNBC, celastrol, mixed micelles, mitochondria targeting, tumor targeting, mitochondrial dysfunction

## Abstract

Considering that the precise delivery of Celastrol (Cst) into mitochondria to induce mitochondrial dysfunction may be a potential approach to improve the therapeutic outcomes of Cst on TNBC, a novel tumor mitochondria dual-targeted mixed-micelle nano-system was fabricated via self-synthesized triphenylphosphonium-modified cholesterol (TPP-Chol) and hyaluronic acid (HA)-modified cholesterol (HA-Chol). The Cst-loaded mixed micelles (Cst@HA/TPP-M) exhibited the characteristics of a small particle size, negative surface potential, high drug loading of up to 22.8%, and sustained drug release behavior. Compared to Cst-loaded micelles assembled only by TPP-Chol (Cst@TPP-M), Cst@HA/TPP-M decreased the hemolysis rate and upgraded the in vivo stability and safety. In addition, a series of cell experiments using the triple-negative breast cancer cell line MDA-MB-231 as a cell model proved that Cst@HA/TPP-M effectively increased the cellular uptake of the drug through CD44-receptors-mediated endocytosis, and the uptake amount was three times that of the free Cst group. The confocal results demonstrated successful endo-lysosomal escape and effective mitochondrial transport triggered by the charge converse of Cst@HA/TPP-M after HA degradation in endo-lysosomes. Compared to the free Cst group, Cst@HA/TPP-M significantly elevated the ROS levels, reduced the mitochondrial membrane potential, and promoted tumor cell apoptosis, showing a better induction effect on mitochondrial dysfunction. In vivo imaging and antitumor experiments based on MDA-MB-231-tumor-bearing nude mice showed that Cst@HA/TPP-M facilitated drug enrichment at the tumor site, attenuated drug systemic distribution, and polished up the antitumor efficacy of Cst compared with free Cst. In general, as a target drug delivery system, mixed micelles co-constructed by TPP-Chol and HA-Chol might provide a promising strategy to ameliorate the therapeutic outcomes of Cst on TNBC.

## 1. Introduction

Triple-negative breast cancer (TNBC) is a highly invasive and intractable breast cancer subtype characterized by a lack of estrogen receptor (ER), progesterone receptor (PR), and human epidermal growth factor receptor 2 (HER 2) [1]. Due to the poor prognosis of TNBC, its annual mortality rate accounts for about 5% of all cancer-related deaths [2]. Currently, TNBC is only moderately responsive to endocrine therapy and some targeted medicine, while traditional chemotherapy often develops resistance, so exploring new therapy methods or new anti-TNBC drugs is still an unmet need.

Celastrol (Cst) is a prospective cancer-specific natural drug extracted from *Tripterygium wilfordii Hook F*, and its anticancer effects on TNBC have been proven to mainly involve mitochondrial dysfunction and mitochondria-mediated apoptosis. Mitochondria are the energy supply factories for cellular life activities, responsible for various biological functions, such as adenosine triphosphate (ATP) synthesis, reactive oxygen species (ROS) production, and apoptosis [3]. In recent years, numerous research studies have shown that mitochondrial function plays a vital role in the development and metastasis of TNBC [4]. Therefore, delivering Cst into the mitochondria to induce mitochondrial dysfunction may be a potential approach to improve the therapeutic outcomes of Cst on TNBC [5].

However, the mitochondrial membrane, composed of a porous outer mitochondrial membrane (OMM) and low-permeability inner mitochondrial membrane (IMM), forms a natural barrier that prevents the passage of exogenous substances, including drug molecules. Fortunately, some exogenous molecules with a delocalized positive charge can enter the IMM and accumulate in mitochondria through electrostatic attraction to the hyperpolarized negative membrane of the mitochondria in cancer cells [6]. Triphenylphosphonium (TPP) is a representative delocalized positively charged ligand with a phosphorus atom in its structure, providing a strong positive charge to electrostatically attract the hyperpolarized negative membrane of mitochondria. Additionally, the three hydrophobic phenyls of TPP can interact with the hydrophobic mitochondrial inner membrane, making it easier to cross the hydrophobic barrier of the mitochondrial membrane [7,8]. Hence, it is often used to construct mitochondrial-targeted delivery systems. Ara et al. developed a novel TPP-functionalized gold nanorod/zinc oxide core–shell nanocomposite (CTPP-GNR@ZnO) for mitochondrial-targeted photothermal therapy and photodynamic therapy [9]. Their research proved that this nanocomposite could target the mitochondria of a murine colorectal carcinoma cell line (CT-26) and remarkably increased the intercellular ROS levels after laser irradiation with 780 nm [10]. Li et al. fabricated mitochondrial-targeting micelles using TPP-modified PEG-PE (1,2-distearoyl-sn-glycero-3-phosphoethanolamine-N-[methoxy(polyethyleneglycol)-2000]) to effectively transport puerarin to the mitochondria and enhance its anti-apoptotic effect [11]. Although the highly positive charge of TPP can be shielded by PEG to promote blood circulation stability, the specific recognition of anti-PEG antibodies to PEGylated nanoparticles is still a noteworthy issue [12].

Furthermore, the enrichment of drugs around tumor cells is a prerequisite for delivering drugs to the mitochondria of tumor cells, which also contributes to obviating the systemic toxicity of some chemotherapeutic drugs [13]. The CD44 receptor is a transmembrane glycoprotein that is specifically highly expressed in breast cancer cells, making it a typical target for identifying tumor cells [14]. Hyaluronic acid (HA) is an acidic mucopolysaccharide characterized by a good biocompatibility, biodegradability, non-immunogenicity, and, in particular, the specific ability to bind with CD44 receptors [15]. Moreover, the unique negative charge of HA confers it the ability to electrostatically attract positively charged nanomaterials, thereby enhancing their blood circulation stability [16,17]. 

In view of this, to accurately deliver Cst into the mitochondria of tumor cells and boost better therapeutic effects on TNBC by inducing mitochondrial dysfunction, novel tumor mitochondria dual-targeting mixed micelles loaded with Cst (Cst@HA/TPP-M) were fabricated via HA-modified cholesterol (HA-Chol) and TPP-modified cholesterol (TPP-Chol), an amphiphilic mitochondrial-targeted material reported in our previous research [18] (Figure 1). HA-Chol was expected to conceal the positive charge of TPP-Chol and recognize tumor cells through CD44 receptors, thereby enabling the mixed micelles to sustain in vivo blood circulation stability and a tumor-targeting ability. After hyaluronidase (HAase) in endo-lysosomes triggered HA degradation, the positive charge of TPP exposed on the mixed micelles would impel endo-lysosomal escape and deliver Cst into the mitochondria in a targeted manner, thereby inducing mitochondrial dysfunction and ultimately killing tumor cells. In this work, the human triple-negative breast cancer cell line MDA-MB-231 with highly expressed CD44 receptors and MDA-MB-231-tumor-bearing nude mice were used to assess the targeting ability and anticancer effectiveness of Cst@HA/TPP-M on TNBC. 

## 2. Materials and Methods

### 2.1. Materials

Cst was provided by Push Biotechnology Co., Ltd. Chengdu, China; Cholesterol, a DCFH-DA fluorescent probe, JC-1 fluorescent probe, and Nile red (NR) were purchased from Solarbio Science & Technology Co., Ltd., Beijing, China; Sodium hyaluronic acid (HA-Na, MW 12 kDa), N, N′-dicyclohexylcarbodiimide (DCC), 4-dimethyl aminopyridine (DMAP), N, N-dimethylformamide (DMF), coumarin 6 (C6), and hyaluronidase (HAase) were obtained from Aladdin Biochemical Technology Co., Ltd., Shanghai, China; Mitotraker red, an ROS detection kit, and an Annexin V-FITC/PI detection kit were supplied by Biyuntian Biotechnology Co., Ltd., Shanghai, China; The MDA-MB-231 cells were bought from the Shanghai Institute of Biochemistry and Cell Biology. High-glucose DMEM, fetal bovine serum (FBS), and trypsin/EDTA solution were supplied by Thermo Fisher Gibico, Waltham, MA, USA. All other chemicals were of analytical or chromatographic grade. BALB/c-nude mice obtained from Changshen Biotechnology Co., Ltd., Liaoning, China, were housed in SPF-grade animal rooms with a 12 h light/dark cycle and fed with standard food and water. All animal experiments were authorized by the Laboratory Animal Ethics Committee of Hubei University of Chinese Medicine.

### 2.2. Synthesis of HA-Chol 

Cholesterol (Chol, 0.382 g), DCC (0.496 g), and DMAP (0.024 g) were dissolved in 20 mL of DMF. Next, DMF solution containing cholesterol, DCC, and DMAP was added dropwise to deionized water containing HA-Na. After 24 h, the reaction solution was put into a dialysis bag with a molecular weight cut-off of 8000–14,000 and dialyzed with distilled water/THF (1:1, *v*:*v*) for three days. Finally, the supernatant liquid in the dialysis bag after centrifugation was freeze-dried to obtain HA-Chol. The structure of HA-Chol was characterized by ^1^H-NMR (Advance 400, Bruker, Fällanden, Switzerland) and FTIR (Vertex 70, Bruker Switzerland). D_2_O was used as an NMR solvent. FTIR detection was performed after dissolving the HA-Chol in distilled water and drying it with a deuterium lamp.

### 2.3. Preparation of Dual-Targeting Mixed Micelles Loaded with Cst (Cst@HA/TPP-M)

Firstly, 9.8 mg of TPP-Chol synthesized by our group was dissolved in 500 µL of anhydrous ethanol and then mixed with 10 mL of deionized water [14]. Next, the mixed solution containing TPP-Chol was dialyzed with distilled water for 24 h. In the meantime, the dialysis medium was replaced three times to completely remove the organic solvent and obtain blank micelles (TPP-M). Afterward, mixed blank micelles (HA/TPP-M) were obtained by mixing TPP-Chol and HA-Chol in an optimal proportion. In total, 2 mg of Cst dissolved in anhydrous ethanol was added to 10 mL of the mixed micelle solution. After stirring for 30 min, the mixed micelle solution containing Cst was dialyzed for 24 h. Finally, the Cst-loaded mixed micelles (Cst@HA/TPP-M) were obtained after centrifugation to remove free Cst. As a control, Cst@TPP-M was prepared according to the same method, just by adding Cst into TPP-M. Micelles loaded with fluorescent reagent C6, or near-infrared reagent NR, were obtained according to the same method as above.

### 2.4. Characterization of Cst@HA/TPP-M

A Malvern particle size meter (Nano ZS90, Malvern, London, UK) was adopted to measure the the particle size, polydispersity (PDI), and Zeta potential of different micelles. The microscopic morphology of the micelles was observed via transmission electron microscopy (TEM-1400, JEOL, Tokyo, Japan) based on a similar negative staining method [19]. Additionally, the entrapment efficiency (EE%) of Cst and the drug loading capacity (DL%) of the mixed micelles were determined in accordance with the low-speed centrifugal method [20]. Briefly, after the micelle solutions were centrifugated with a speed of 1006× *g* for 10 min, the content of Cst in the micelle solution before and after centrifugation was determined using a High-Performance Liquid Chromatograph (HPLC, 1260 Infitiny, Agilent) with an InertSustain C18 chromatographic column (4.6 mm × 250 mm, 5 μm). The mobile phase consisted of 1% glacial acetic acid and methanol with a volume ratio of 15:85 and a flow rate of 1.0 mL/min while maintaining the temperature at 30 °C. The injection volume was set to 20 μL and the UV detection wavelength was set at 425 nm. The EE% and DL% were computed using the following formulas:(1)$$\mathrm{EE}\;(\%)=\;\frac{\mathrm{Weight}\;\mathrm{of}\;\mathrm{encapsulated}\;\mathrm{Cst}\;\mathrm{in}\;\mathrm{micelles}}{\mathrm{Total}\;\mathrm{weight}\;\mathrm{of}\;\mathrm{Cst}}\times 100\%$$
(2)$$\mathrm{DL}\;(\%)=\;\frac{\mathrm{Weight}\;\mathrm{of}\;\mathrm{encapsulated}\;\mathrm{Cst}\;\mathrm{in}\;\mathrm{micells}}{\mathrm{Weight}\;\mathrm{of}\;\mathrm{total}\;\mathrm{micelles}\;+\;\mathrm{Weight}\;\mathrm{of}\;\mathrm{encapsulated}\;\mathrm{Cst}\;\mathrm{in}\;\mathrm{micells}}\times 100\%$$

### 2.5. Stability of the Micelles

The Cst-loaded micelles were stored at 4 °C for a varying amount of time to evaluate their low-temperature storage stability. In addition, to evaluate their serum stability, Cst@HA/TPP-M and Cst@TPP-M were separately mixed with 10% FBS (1:1, *v*:*v*) to a Cst concentration of 0.5 mg/mL, and then incubated at 37 °C with shaking for 24 h. Changes in particle size, Zeta potential, or EE% over time were determined.

### 2.6. HA Degradation of Cst@HA/TPP-M

According to Xiao’s method [18], HA degradation mediated by HAase was surveyed by monitoring the Zeta potential of Cst@HA/TPP-M incubated with or without HAase (1 mg/mL) at pH 7.4 and pH 5.0 over time.

### 2.7. In Vitro Drug Release of the Micelles

The drug release characteristics of the different micelles were evaluated via the same dialysis method reported in the literature [21]. Briefly, Cst@HA/TPP-M solution with a 50 μg/mL Cst concentration was placed in a dialysis bag (MWCO 3500) and then soaked in 20 mL of release medium composed of pH 7.4 phosphate-buffered solution (PBS) and 1% Tween 80. In total, 1 mL of the release medium was collected at pointed times, such as 1, 2, 4, 8, 12, 24, 48, and 72 h. Meanwhile, the release medium with the same volume and temperature was replenished to ensure that the dialysis environment remained unchanged. The Cst content in the collected release medium was detected using HPLC according to the same conditions as previously described. As controls, the release behaviors of Cst@TPP-M and free Cst were also investigated according to the same method above.

### 2.8. Hemolysis Test

Fresh rabbit blood was adopted to prepare a 2% erythrocyte suspension. A total of 200 μL of erythrocyte suspension was mixed with 800 μL of saline, Triton X-100 solution, and micellar solutions with different Cst concentrations as a negative group, positive group, and test group, respectively. After incubation at 37 °C for 2 h, the mixed solution was centrifuged for 10 min to take away intact erythrocyte, and the OD value of the supernatant at 570 nm was detected by a microplate reader (x-Mark, Bio-Rad, Hercules, CA, USA) [22]. The hemolysis percentage was calculated according to the following method.
(3)$$\mathrm{Hemolysis}\;\mathrm{percentage}=\frac{\mathrm{ODtest}\;-\;\mathrm{ODneg}}{\mathrm{ODpos}\;-\;\mathrm{ODneg}}\;\times 100\%$$

ODtest, ODneg, and ODpos represent the absorbance values at 570 nm of the test group, negative group, and positive group, respectively.

### 2.9. In Vitro Cellular Uptake

MDA-MB-231 cells were seeded into a 6-well plate at a density of 2.0 × 10^5^ cells/well. After cell adhesion, C6 and C6@HA/TPP-M solutions with a C6 concentration of 100 ng/mL were added to the 6-well plates and incubated with the cells for another 4 h. After being washed with PBS, the cells were stained with Hoechst33342 and photographed with an inverted fluorescence microscope (CKX31, Olympus, Tokyo, Japan). Flow cytometry (ACEA NovoCyte, Agilent, Santa Clara, CA, USA) was used to quantitatively analyze the fluorescence intensity of C6 in the cells. Moreover, the MDA-MB-231 cells were pretreated with excess HA for 1 h before treatment with C6@HA/TPP-M to further evaluate the effect of the CD44 receptor on the cellular uptake amount of C6.

### 2.10. Endo-Lysosomal Escape and Mitochondrial Localization Study

After adhesion and growth in the confocal laser dish for 24 h, MDA-MB-231 cells were exposed to C6@HA/TPP-M for 1, 2, and 4 h, respectively. The incubation concentration of C6 was 100 ng/mL. Subsequently, the cells were stained with Lyso-Tracker Red and photographed by a laser confocal microscope (IX81-FV1000, Olympus, Pine Brook, NJ, USA).

The MDA-MB-231 cells cultured in the confocal laser dish were incubated with free C6 and C6@HA/TPP-M for 4 h. After being washed with PBS, the cells were dyed with Mito-Tracker Red CMXRos solution for 30 min and photographed by a laser confocal microscope.

### 2.11. ROS Assay

A DCFH-DA fluorescent probe was utilized to determine the ROS levels of the MDA-MB-231 cells after incubation with different Cst-loaded micelles. After adhesion and growth in 6-well plates, the cells were incubated with free Cst and Cst@HA/TPP-M at a Cst concentration of 0.5 µg/mL for 24 h, respectively. Next, after being washed by PBS, the cells were digested, collected, and stained with DCFH-DA for 20 min at 37 °C. The cells were then washed with serum-free medium and resuspended with PBS. Finally, the intracellular fluorescence was examined via an inverted fluorescence microscope and the intracellular fluorescence intensity was quantified through flow cytometry.

### 2.12. Mitochondrial Membrane Potential Study

The mitochondrial membrane potential difference was surveyed via a fluorescent probe JC-1 after MDA-MB-231 cells were treated with different Cst-loaded micelles. After adhesion and growth in a 6-well plate, the cells were incubated with free Cst and Cst@HA/TPP-M at a Cst concentration of 0.5 µg/mL for 24 h, respectively. Next, after removing the free Cst and Cst@HA/TPP-M, the cells were treated with the JC-1 kit. Similarly, flow cytometry was used to quantify the intracellular fluorescence intensity.

### 2.13. Cell Apoptosis Assay

The apoptosis induced by free Cst and Cst@HA/TPP-M in the MDA-MB-231 cells was evaluated via the Annexin V-FITC/PI staining method. Briefly, after adhesion and growth in 6-well plates, the cells were incubated separately with free Cst and Cst@HA/TPP-M (the Cst concentration was 1 μg/mL) for 24 h, and then the cells were washed, collected, and resuspended in a binding buffer. Prior to flow cytometry analysis, the cells were sequentially stained with Annexin V-FITC and PI according to the operation instructions of the Annexin V-FITC/PI kit.

### 2.14. Cell Viability Assay

The cytotoxicity of Cst@HA/TPP-M on the MDA-MB-231 cells and blank micelles (TPP-M and HA/TPP-M) on LO2 human normal liver cells was tested using the MTT method [23]. Briefly, after adhesion and growth in 96-well plates for 24 h, the cells were incubated with different micelle solutions for an additional 24 h. After that, the cells were further cultured with MTT solution for 4 h. The formazan crystals in each well were dissolved using DMSO, and the OD value of each well at 570 nm was detected with an ELISA reader (x-Mark, Bio-Rad, Hercules, CA, USA).

### 2.15. In Vivo Imaging Investigation

In total, 0.2 mL of MDA-MB-231 cell suspension with a concentration of 5 × 10^7^ cells per milliliter was transplanted to the right breast pad of female BALB/c nude mice aged from 4 to 5 weeks. When the tumor grew to about 100 mm^3^ in diameter, the MDA-MB-231-tumor-bearing nude mice were separated into three groups at random (*n* = 5 for each group) and treated with 200 μL of NR@TPP-M, NR@HA/TPP-M, and free NR (the NR concentration was 0.04 mg/mL) via the caudal veins, respectively. At 1 h, 3 h, 6 h, 12 h, and 24 h after treatment, the NR signal in the mice was monitored through in vivo imaging. After 24 h of observation, the mice were euthanized, and their hearts, livers, spleens, lungs, kidneys, and tumors were dissociated and observed via in vivo imaging.

### 2.16. In Vivo Antitumor Efficacy

An MDA-MB-231 xenograft tumor model in female BALB/c nude mice was constructed using the same method as the in vivo imaging experiment. When the tumor volume developed to about 100 mm^3^, the mice were separated into three groups at random (*n* = 6 for each group) and treated with saline, free Cst, and Cst@HA/TPP-M via a caudal vein injection every other day, respectively. The injection dose of Cst in the treatment group was 2 mg/kg, and the treatment proceeded for 14 days. During the treatment, the tumor volumes and body weights of the mice were closely monitored.

### 2.17. Statistical Analysis

All experimental data were reported as mean ± standard deviation (SD) and analyzed via Prism software 10.0 version (GraphPad Inc., NewYork, NY, USA). Statistical comparisons were performed through Student’s *t*-test and one-way ANOVA. When * *p* < 0.05, ** *p* < 0.01, and *** *p* < 0.001, it was considered to have statistical differences.

## 3. Results and Discussion

### 3.1. Synthesis and Characterization of HA-Chol

HA-Chol was synthesized by esterification between cholesterol and hyaluronic acid [24]. The chemical structure of HA-Chol was confirmed by hydrogen spectroscopy and Fourier transform infrared spectroscopy (Appendix A), and the graft rate of cholesterol on HA was estimated at 30% according to the hydrogen spectroscopy.

### 3.2. Preparation and Characterization of Mixed Micelles

The mixed micelles were prepared by co-solvent precipitation, and anhydrous ethanol was selected as an organic solvent. Firstly, the molar ratios of TPP-Chol and HA-Chol were screened, and the results showed that only when the molar ratio of TPP-Chol and HA-Chol was 1:1 was the entrapment efficiency of Cst higher than 90%, with a smaller particle size (133.7 ± 3.7 nm) and less than 0.2 of PDI (Appendix A), displaying the effective load of the mixed micelles on Cst. Additionally, the stirring speeds and time during the preparation of the mixed micelles were also optimized, with the results shown in Appendix A. Under the optimal conditions of a molar ratio of 1:1, stirring speed of 300 revolutions per minute, and stirring time of 60 min, the particle size of Cst@HA/TPP-M was 131.4 ± 2.4 nm, the Zeta potential was −19.7 ± 0.6 mV, the Cst entrapment efficiency exceeded 97%, and the drug loading capacity was up to 22.8% (Table 1). By comparison, the particle size of Cst@TPP-M was smaller at only 94.6 ± 2.2 nm, but the potential was strongly positive, up to 38 mV, reflecting that the addition of HA-Chol could shield the positive charge of TPP-Chol and increase the particle size of the mixed micelles. Interestingly, the entrapment efficiency of HA/TPP-M was higher than that of TPP-M. This might have been because the addition of HA-Chol increased the hydrophobic region of HA/TPP-M, thereby allowing HA/TPP-M to load more Cst. The Cst@HA/TPP-M solution was clear and an obvious Tyndall effect appeared (Figure 1A). Under TEM observation, Cst@HA/TPP-M was nearly spherical and showed distinct core–shell structures (Figure 1B).

### 3.3. Stability Studies

After being stored at a low temperature for 20 days, the particle size of Cst@HA/TPP-M did not change significantly (Figure 1C) and the entrapment efficiency of Cst remained above 90% (Figure 1D), while the particle size and entrapment efficiency of Cst@TPP-M also did not show significant changes, indicating that Cst@HA/TPP-M and Cst@TPP-M both had a certain low-temperature storage stability. However, the particle size of Cst@TPP-M dramatically increased from 97.3 ± 1.2 nm to 128 ± 2.4 nm after incubation with 10% FBS (Figure 1E), and the potential became negative after 12 h of incubation (Figure 1F), indicating that the serum stability of Cst@TPP-M was relatively poor. This may have been because positively charged Cst@TPP-M can combine with some negatively charged substances in serum (such as serum proteins), leading to charge reversal and a particle size increase. This result proved that HA-Chol was conducive to improving the stability of the mixed micelles during in vivo blood circulation.

### 3.4. Degradation of HA

It has been reported that HA can be degraded by highly activated HAase in endo/lysosomes [25]. Also, pH and ionic strength are able to affect the hyaluronan hydrolysis catalyzed by HAase [26]. Hence, Cst@HA/TPP-M was incubated with HAase under acidic and neutral conditions to investigate whether HA degradation changed the potential of Cst@HA/TPP-M. As presented in Figure 2A, after cultivation at pH 7.4 for 6 h, the potential of Cst@HA/TPP-M changed from −19 mV to about −10 mV, whereas the potential became positive at pH 5.0. Notably, the potential of Cst@HA/TPP-M changed more rapidly in the presence of HAase, suggesting that the acidic environment and HAase could expedite HA degradation. Therefore, it could be inferred that, when Cst@HA/TPP-M entered acidic and HAase-rich lysosomes, HA would be rapidly degraded, and then TPP would be exposed, resulting in the potential converse of mixed micelles to positive charge.

### 3.5. In Vitro Drug Release

As presented in Figure 2B, the free Cst group released over 90% of their Cst within 12 h, suggesting rapid drug release behavior. However, Cst@TPP-M only released about 18% of Cst, while the release amount of Cst@HA/TPP-M within 12 h was approximately 30%. Even after 72 h, the cumulative release amount of Cst in the micelle groups was still less than 40%, indicating that the Cst-loaded micelles displayed sustained drug release properties.

### 3.6. Safety Evaluation

Considering that the strong positive charge of TPP may damage red blood cells [27], the hemolysis rates of Cst@HA/TPP-M and Cst@TPP-M were evaluated. As shown in Figure 2C, the hemolysis rate of Cst@HA/TPP-M was lower than that of Cst@TPP-M in all measured concentration ranges, showing significant statistical differences (*p* < 0.01). Especially when the concentration of Cst was higher than 0.5 mg/mL, the hemolysis rate of Cst@TPP-M exceeded 5%, indicating that Cst@TPP-M had a certain hemolytic effect and might cause red blood cell damage during in vivo blood circulation. Conversely, after adding HA-Chol to form mixed micelles, the hemolysis rate was less than 3% in all measured concentration ranges, suggesting that HA-Chol could improve the in vivo safety of the mixed micelles due to its electrostatic attraction with TPP.

Due to the hemolytic effect of Cst@TPP-M, the cytotoxicity of the blank micelles to the LO2 cells was detected to investigate whether HA-Chol can reduce the toxicity of TPP and increase the biosafety of micelles. LO2 cells are a commonly used cell model for evaluating liver toxicity, typically used to assess the safety or toxicity of exogenous biological materials [28]. As shown in Figure 2D, after incubation for 24 h, TPP-M led to a higher cytotoxicity on the LO2 cells compared with HA/TPP-M. When the concentration of TPP-Chol was 200 μg/mL, TPP-M exhibited obvious cytotoxicity, causing over 40% cell death, while HA/TPP-M produced a similar cytotoxicity only when the concentration of TPP-Chol reached 800 µg/mL. But, at this concentration, the Cst concentration reached 160 µg/mL, far higher than the maximum experimental dose of 10 µg/mL in the subsequent cell experiments, suggesting that the electrostatic interaction of HA-Chol with TPP-Chol successfully mitigated the toxicity induced by TPP. According to previous experimental results, it can be concluded that TPP-Chol could self-assemble into micelles and load Cst, but its stability and safety were weaker than the mixed micelles co-assembled by HA-Chol and TPP-Chol.

### 3.7. Cellular Uptake

Liposoluble probe C6 with green fluorescence is a commonly used alternative to liposoluble drugs without fluorescence for the qualitative and quantitative observation of cellular uptake [29]. Because the particle size and Zeta potential of C6@HA/TPP-M were close to those of Cst@HA/TPP-M (Table 1), it was feasible to replace Cst with C6. It was observed that a large amount of C6 green fluorescence appeared around the blue cell nuclei after incubation with C6@HA/TPP-M for 4 h, whereas a small amount of green fluorescence appeared in the free-Cst-treated cells (Figure 3A). Flow cytometry analysis verified that the fluorescence intensity of C6@HA/TPP-M was three times higher than that in the free C6 group (Figure 3B,C), implying that HA/TPP-M has the potential to deliver drugs into cells and efficiently elevate the drug uptake amount in cells. To further prove that the increasing C6 uptake amount in the C6@HA/TPP-M group may have been related to the CD44 receptors highly expressed on the cytomembrane, MDA-MB-231 cells were pretreated with excessive HA for 1 h before treatment with C6@HA/TPP-M. The result showed that the fluorescence intensity declined sharply, which was significantly different from C6@HA/TPP-M without HA pretreatment (*p* < 0.001). The preferential binding of excessive HA and CD44 receptors [30] might have caused a decrease in receptors that could bind C6@HA/TPP-M. According to this result, it can be inferred that the cell uptake mechanism of Cst@HA/TPP-M might depend on the CD44-receptor-mediated endocytosis pathway.

### 3.8. Endo-Lysosomal Escape and Mitochondrial Localization

Jayanth et al. reported that biodegradable nanoparticles consisting of poly (DL-lactide-*co*-glycolide) can selectively reverse the surface charge of nanoparticles from negative to positive in the acidic environment of endo-lysosomes, thus quickly escaping endo-lysosomes into the cytosol [31]. Our previous experiment proved that the acidic environment was an important factor for the degradation of HA, and the potential of Cst@HA/TPP-M would be changed from negative to positive after HA degradation. Therefore, C6@HA/TPP-M might easily escape endo-lysosomes due to the charge imbalance in endo-lysosomes caused by HA degradation [32]. As illustrated in Figure 4A, after incubation with C6@HA/TPP-M for 1 h, the green fluorescence presented by C6 mainly coincided with the red fluorescence presented by the endo-lysosomes in the cells, producing a remarkable yellow fluorescence, which indicated that C6@HA/TPP-M might have already entered the endo-lysosomes. However, after 2 h, this yellow fluorescence gradually weakened, and after 4 h, a faint green fluorescence appeared, meaning that C6@HA/TPP-M might escape from endo-lysosomes and enter the cytoplasm. The cell nuclei in Figure 4B were stained blue by Hoechst33342, and the mitochondria were stained red. Compared to the weaker green fluorescence of the free C6 group, C6@HA/TPP-M exhibited a stronger green fluorescence around the labeled blue nuclei, indicating an increase in C6 intake, which was consistent with the previous experimental result of cell uptake. Notably, the yellow fluorescence representing the overlapping of C6 and mitochondria obviously appeared around the blue nuclei of the C6@HA/TPP-M group, suggesting that the mixed micelles constructed by TPP-Chol facilitated the mitochondria-targeted delivery of C6.

### 3.9. ROS Level

Porporato et al. [33] reviewed that excessive levels of ROS not only promote the migration and invasion of TNBC cells, but also result in ROS-induced apoptosis and damage. For survival, mitochondria cleverly maintain the level of ROS. Therefore, emerging excessive ROS generation to destroy the balance maintained by the mitochondria may be a good way to induce TNBC cell death. After the same incubation time, the ROS levels of free Cst and Cst@HA/TPP-M were detected using an ROS detection kit. As presented in Figure 5A–C, the fluorescence intensity of Cst@HA/TPP-M was evidently higher than that of the free Cst group (*p* < 0.001), suggesting that Cst@HA/TPP-M effectively increased the ROS level, which might have resulted from the increased drug uptake and successful mitochondria-targeted delivery proven by the previous experiments.

### 3.10. Mitochondrial Membrane Potential

Mitochondrial damage can lead to mitochondrial depolarization and a decrease in membrane potential, an essential index for evaluating mitochondrial dysfunction [34]. JC-1 is a novel cationic carbonyl cyanine dye that can detect changes in mitochondrial membrane potential through the red–green fluorescence produced by JC-1 at different mitochondrial membrane potentials. As shown in Figure 5D, after the MDA-MB-231 cells were treated with Cst@HA/TPP-M, the ratio of red and green fluorescence decreased dramatically, obviously lower than that of the free Cst group (*p* < 0.001), implying that Cst@HA/TPP-M could effectively reduce the membrane potential and induce mitochondrial depolarization. Combined with the previous ROS level result, it is reasonable that Cst@HA/TPP-M successfully induced mitochondrial dysfunction after the targetid delivery of Cst to the mitochondria.

### 3.11. Cell Apoptosis

Increasing evidence shows that the antitumor activity of Cst on TNBC is related to its apoptosis-inducing effect [5], so Annexin V/PI double staining was employed to investigate whether Cst@HA/TPP-M could promote the apoptosis-inducing effect of Cst on MDA-MB-231 cells. As expected, the total apoptosis rate of Cst@HA/TPP-M was about 40%, far higher than the 24% in the free Cst group (Figure 6A,B). This result certified that Cst could better exert its pro-apoptotic activity after being entrapped by mixed micelles.

### 3.12. Cell Viability

The results, as illustrated in Figure 6C, revealed that Cst@HA/TPP-M was more efficient than the free Cst in inducing MDA-MB-231 cell death. According to a cell viability analysis, the IC_50_ of Cst@HA/TPP-M was 1.162 μg/mL, about half the IC_50_ value of the free Cst group. The better inhibitory effect of Cst@HA/TPP-M on the growth of MDA-MB-231 cells might be attributed to a higher Cst uptake, an effective imbalance of ROS levels, and more cell apoptosis.

### 3.13. In Vivo Imaging

Nile red is a lipophilic near-infrared dye with orange–red fluorescence, commonly used as a substitute for non-fluorescent lipophilic drugs for in vivo imaging observation [32]. After NR was entrapped by the mixed micelles, the particle size and Zeta potential of NR@HA/TPP-M were close to that of Cst@HA/TPP-M (Table 1), indicating that NR@HA/TPP-M can replace Cst@HA/TPP-M to observe the in vivo distribution of drugs. According to Figure 7A, it can be found that the NR signal of NR@HA/TPP-M mainly appeared at the tumor site under the right breast pad after 3 h from injection, and was still clear 12 h later. In contrast, the NR signal of NR@TPP-M partially accumulated in the bladder within 3 h after injection, implying a possible rapid clearance of NR. Figure 7B showed that the radiant efficiency in the tumor site of the NR group obviously declined after 3 h compared with NR@HA/TPP-M and NR@TPP-M. After 24 h, the NR signal of the NR@HA/TPP-M group on the tumor was significantly stronger than that of the NR group (*p* < 0.001) and NR@TPP-M (*p* < 0.001), (Figure 7C,D). These results indicated that NR@HA/TPP-M has the potential to promote the long-term accumulation of drugs at the tumor site.

### 3.14. In Vivo Pharmacodynamics Study

As shown in Figure 8A, the weight of the mice in the free Cst group began to decrease from the 18th day after tumor inoculation, completely different from the continuous increase in the body weight of the mice in the saline and Cst@HA/TPP-M groups during treatment. Our previous research also found this weight loss phenomenon in mice during treatment with Cst [35], which was believed to be due to side effects such as a poor appetite due to the systemic distribution of free Cst in the mice. The tumor volume of the free Cst group continued to increase, exceeding that of Cst@HA/TPP-M after 14 days from injecting tumor cells, but the growth rate of the tumor volumes was slower than that of the saline group (Figure 8B, *p* < 0.01). Comparatively, Cst@HA/TPP-M displayed a superior therapeutic efficacy against TNBC in mice, with a tumor inhibitory rate of about 55%, significantly higher than the 35% of the free Cst group (Figure 8D, *p* < 0.001). The tumor index of Cst@HA/TPP-M was lower than that of the free Cst group (Figure 8C, *p* < 0.01), and its tumor size was also relatively small (Figure 8E). These results consistently demonstrate that Cst@HA/TPP-M contributed to better inhibition of the development of TNBC in the mice and a reduction in the toxic side effects of Cst.

## 4. Conclusions

In summary, Cst@HA/TPP-M mixed micelles with a tumor mitochondria dual targeted ability were prepared to induce mitochondrial dysfunction and upgrade the therapeutic efficacy of Cst on TNBC. The research results showed that Cst@HA/TPP-M decreased the hemolysis rate, improved the in vivo stability and safety, and promoted the long-term accumulation of drugs at the tumor site compared with Cst@TPP-M. Cst@HA/TPP-M not only increased the cellular uptake of drugs through CD44-mediated internalization, but also successfully achieved endo-lysosomal escape and mitochondrial-targeting transport through charge reversal after HA degradation, thereby significantly increasing the ROS levels in the MDA-MB-231 cells, reducing the mitochondrial membrane potential, and promoting cell apoptosis. Notably, compared to free Cst, Cst@HA/TPP-M significantly inhibited the development of TNBC in mice and avoided weight loss during treatment. Overall, the mixed micelles co-constructed by TPP-Chol and HA-Chol may be a promising targeted delivery nanocarrier to improve the therapeutic potential of Cst for TNBC.

## Data Availability

Datasets used and analyzed during the current study are available from the corresponding author upon reasonable request.

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
