# Peer review of "Development of Dual-Targeted Mixed Micelles Loaded with Celastrol and Evaluation on Triple-Negative Breast Cancer Therapy"

_pharmaceutics, 2024, doi:10.3390/pharmaceutics16091174_

Round 1
Reviewer 1 Report (Previous Reviewer 4)
Comments and Suggestions for Authors
The manuscript "The development of dual-targeted mixed micelles loaded with celastrol and evaluation on triple-negative breast cancer therapy" is a well-documented study. I commented the earlier version of this work and the authors have taken into account my recommendations. I can see now only some language problems (see below).
Comments on the Quality of English LanguageSome non-standard expressions persist e.g (section 2.5) "Cst-loaded micelles were stored at 4℃ for different days to evaluate low-temperature storage stability." Probably, "for different days" it is better to replace with "for a varying amount of time". Also, some long sentences are difficult to understand.
Author Response
The manuscript "The development of dual-targeted mixed micelles loaded with celastrol and evaluation on triple-negative breast cancer therapy" is a well-documented study. I commented the earlier version of this work and the authors have taken into account my recommendations. I can see now only some language problems (see below).
Some non-standard expressions persist e.g (section 2.5) "Cst-loaded micelles were stored at 4℃ for different days to evaluate low-temperature storage stability." Probably, "for different days" it is better to replace with "for a varying amount of time". Also, some long sentences are difficult to understand.
R: Thank you for your suggestion! We have changed "for different days" to "for a varying amount of time" in section 2.5. We also have checked and revised the long sentences in the manuscript, and the modified parts have been highlighted in red.

Reviewer 2 Report (Previous Reviewer 1)
Comments and Suggestions for Authors
In the manuscript, the authors describe new micellar system for mitochondria targeting and improved drug delivery to cancer cells. The experiments were soundly planned and the results accompanied by adequate discussion. Some minor points should be addressed to improve the quality of the manuscript.
1. Introduction, 3rd paragraph: triphenylphosphonium is a positively charged species, while triphenylphosphine is not
2. Section 2.3: Please, indicate the feed amont of Cst
3. Section 2.4: Since rpm is rotor-dependent, please, provide rcf value to be more informative. Unincorporated Cst was quantified in the supernatant
4. Section 2.5: For the serum stability experiment, were the micelles diluted in 10% FBS? If so, to what concentration? The same as freshly prepared micelles?
5. Section 2.7: What was the composition of micellar solution used for release experiments? Freshly prepared micelles or micelles containing some predefined drug concentration?
6. Section 2.13: Please, indicate Cst concentration used in experiment.
7. Section 2.15: Please, provide the number of animals per group. Also, the concentration of NR.
8. Section 2.16: Should it be MDA-MB-231 instead of A-MB-231?
9. Section 3.1: Although FTIR and 1H NMR were used to characterize prepared polymers, they were not described in Methods. Please, include experimental conditions or provide adequate reference.
10. In Tables S2 and S3, please indicate the stirring time and stirring speed in Table captions.
11. Section 3.10: Should Figure 5D be referenced in this Section?
12. Section 3.14: When describing tumor volume in the text, should the referred figure be 8B instead of 8A?
Author Response
In the manuscript, the authors describe new micellar system for mitochondria targeting and improved drug delivery to cancer cells. The experiments were soundly planned and the results accompanied by adequate discussion. Some minor points should be addressed to improve the quality of the manuscript.
- Introduction, 3rdparagraph: triphenylphosphonium is a positively charged species, while triphenylphosphine is not.
R: Thank you for your comments! We have revised the description of TPP in the introduction 3rd paragraph.
- Section 2.3: Please, indicate the feed amount of Cst
R: Thank you for your comments! We added the feed amount of Cst in section 2.3.
- Section 2.4: Since rpm is rotor-dependent, please, provide rcf value to be more informative. Unincorporated Cst was quantified in the supernatant
R: Thank you for your comments! We have provided rcf value and revised formula in section 2.4.
- Section 2.5: For the serum stability experiment, were the micelles diluted in 10% FBS? If so, to what concentration? The same as freshly prepared micelles?
R: Thank you for your comments! We have added the detailed operation method and the drug concentration in section 2.5.
- Section 2.7: What was the composition of micellar solution used for release experiments? Freshly prepared micelles or micelles containing some predefined drug concentration?
R: Thank you for your comments! We have added the composition of micellar solution and drug concentration in section 2.7.
- Section 2.13: Please, indicate Cst concentration used in experiment.
R: Thank you for your comments! We have added the drug concentration used in section 2.13.
7.Section 2.15: Please, provide the number of animals per group. Also, the concentration of NR.
R: Thank you for your comments! We have added the number of animals and the concentration of NR in section 2.15.
8.Section 2.16: Should it be MDA-MB-231 instead of A-MB-231?
R: Thank you for your advice! We are very sorry for the word error and have corrected it.
- Section 3.1: Although FTIR and 1H-NMR were used to characterize prepared polymers, they were not described in Methods. Please, include experimental conditions or provide adequate reference.
R: Thank you for your comments! We have added the detailed method in section 2.2.
- In Tables S2 and S3, please indicate the stirring time and stirring speed in Table captions.
R: Thank you for your advice! We have indicated the stirring time and stirring speed in Table captions.
- Section 3.10: Should Figure 5D be referenced in this Section?
R: Thank you for your comments! We have referenced Figure 5D in section 3.10.
- Section 3.14: When describing tumor volume in the text, should the referred figure be 8B instead of 8A?
R: Thank you for your advice! We are very sorry for the word error and have corrected it.

Reviewer 3 Report (New Reviewer)
Comments and Suggestions for Authors
The manuscript aim to propose a dual-targeted mixed micelles loaded with celastrol with an evaluation on triple-negative breast cancer therapy.
On the Materials and methods section, it is claimed the following: “Next, after being washed by PBS, the cells were treated with ROS kit following the instructions of ROS kit.” The precise ROS detection kit used cannot be retrieved: please be specific, as it looks not the most worldwide used (hence comparable and reproducible).
Figure 3 looks remarkably "shrinked" and difficult to understand. Please, amend.
Figure 4 reports on a very poor resolution, with the caption on the following page. Please, provide for amendment.
Moreover, references list must be improved with additional, broader state-of-the-art sources, comparing different PDT agents and triggers for their delivery, in order to enhance the core discussion (e.g. doi: 10.3389/fchem.2020.573211), mainly in the introduction section.
It is suggested to run an English check throughout the whole manuscript (e.g. “More than that, the unique negative charge of HA can endow it with the ability to shield positively charged nanomaterials, thereby improving in vivo transport stability of nanomaterials[13,14].
In view of this,”).
The conclusion section is actually very poor and does not report on all findings in a thorough way. It might also help discuss over an additional section any possible limitations/future perspectives.
Comments on the Quality of English LanguageIt is suggested to run an English check throughout the whole manuscript (e.g. “More than that, the unique negative charge of HA can endow it with the ability to shield positively charged nanomaterials, thereby improving in vivo transport stability of nanomaterials[13,14].
In view of this,”).
Author Response
The manuscript aims to propose a dual-targeted mixed micelles loaded with celastrol with an evaluation on triple-negative breast cancer therapy.
On the Materials and methods section, it is claimed the following: “Next, after being washed by PBS, the cells were treated with ROS kit following the instructions of ROS kit.” The precise ROS detection kit used cannot be retrieved: please be specific, as it looks not the most worldwide used (hence comparable and reproducible).
R: Thank you for your comments! We have added the method in section 2.11.
Figure 3 looks remarkably "shrinked" and difficult to understand. Please, amend.
R: Thank you for your advice! We have revised Figure 3.
Figure 4 reports on a very poor resolution, with the caption on the following page. Please, provide for amendment.
R: Thank you for your advice! We have revised Figure 4.
Moreover, references list must be improved with additional, broader state-of-the-art sources, comparing different PDT agents and triggers for their delivery, in order to enhance the core discussion (e.g. doi: 10.3389/fchem.2020.573211), mainly in the introduction section.
R: Thank you for your comments! We have added the discussion in the introduction section and updated the references list.
It is suggested to run an English check throughout the whole manuscript (e.g. “More than that, the unique negative charge of HA can endow it with the ability to shield positively charged nanomaterials, thereby improving in vivo transport stability of nano materials[13,14].In view of this,”).
R: Thank you for your comments! We have checked and modified the English throughout the whole manuscript.
The conclusion section is actually very poor and does not report on all findings in a thorough way. It might also help discuss over an additional section any possible limitations/future perspectives.
R: Thank you for your suggestions! We have revised the conclusion section.

This manuscript is a resubmission of an earlier submission. The following is a list of the peer review reports and author responses from that submission.
Round 1
Reviewer 1 Report
Comments and Suggestions for Authors
The manuscript by Huang et al describes the preparation and formulation optimization, in vitro biological activity, and bioavailability of stimuli-sensitive hyaluronic acid micelles for mitochondrial targeting. Hyaluronic acid is a highly attractive drug carrier and the authors have used an array of techniques for detailed characterization, and to obtain relevant information about this system. Although the manuscript possesses the merits to be published in this journal, some details need to be clarified:
1. In Scheme 1, replace DAMP with DMAP
2. Introduction: please, provide the full name for PEG-PE (page 2, bottom). Also, provide the complete name of Tripterygium. Is it the whole genus, or just a certain species that possesses medicinal properties?
3. In Section 2.3, please, provide the reference for TPP-Chol synthesis. Please, provide the method and the conditions used for DL and EE.
4. Section 2.6: I am wondering how this experimental setup works. If the micellar solution is packed in a membrane with a cutoff of 3.5kDa which is immersed in a medium containing HAase (50k Da) as described, how does it ever get exposed to and degraded by HAase?
5. Section 2.7: please, provide experimental conditions for HPLC.
6. Section 2.9 and 2.15: Please, describe the preparation of micelles loaded with C6 and NR.
7. Section 2.15: please, provide information on how the animals were sacrificed. Also, describe how the organs were dissociated and observed.
8. Table S1: The optimization of producing parameters is relevant from a technological point of view. Could you provide the influence of stirring speed and time on micellar parameters similarly as presented in Table S1?
9. In FigureS1 A, C it is difficult to observe the individual peaks. Since there are no peaks above 5ppm it would be beneficial to zoom in on the region of the peaks.
10. In Figure 1F the influence of FBS on the zeta potential is presented (changing from positive to negative in the presence of FSB in the case of TPP-M). However, when commenting on the increased uptake (see Fig 3) of TPP-M, the authors justify it with an increased positive charge. Could you comment on it? Was the complete medium used in in vitro experiments?
11. One of the major points is the absence of negative controls. Did the authors measure the cytotoxic effects of blank micelles?
12. Figure 7: could you provide an alternative type of presentation of ex vivo fluorescence? It is a bit difficult to quantitatively evaluate accumulation just by looking at the presented figures.
Reviewer 2 Report
Comments and Suggestions for Authors
The manuscript submitted by Huang et al. deals with the targeted delivery of Celastrol to TNBC cells using mixed micelles able to target mitochondria and causing mitochondrial dysfunction.
The experimental work was very well planned and has considerable importance in the field of research. The authors provide a systematic approach to handle all key issues associated with drug delivery using micelles and clearly offer a comprehensible results section. Overall the manuscript is very well written.
Major comments:
Essentially two major comments can be made to the work that need consideration by the authors:
i) The authors do not provide a true discussion section of their interesting results. Results are well presented but a meaningful discussion is essentially missing. The manuscript would be much improved by handling this issue.
ii) The studies involving tumours in nude mice made use of Nile Red (NR) instead of Cst to monitor tumor-targeting ability by the developed micelles but unfortunately did not monitor the antitumoral effect of Cst loaded micelles. This presentation of such result would have been the “ultimate proof of concept” of the efficacy of the dual-targeted mixed micelles.
Reviewer 3 Report
Comments and Suggestions for Authors
The significant issue with this work is the lack of information regarding the number of biological repetitions performed and the lack of standard deviations on some graphs, suggesting that the experiment was conducted only once.
Does using coumarin instead of Cst change the particle size or physicochemical properties?
If I understand correctly, the Authors used as control micelles with Cst (there is no information about the solvent used to dissolve Cst); however, as a control for all experiments, the Authors should use two additional controls: empty micelles and cell culture medium to evaluate if the micelles can affect the cells' viability.
Did the Authors use MTT at a concentration of 5 mg/mL since the standard final concentration of MTT in cell culture is approx 0.5 mg/mL?
Did the Authors inject 50 million cells into each mouse? Which solvent was used for the cancer cell implantation? Could the Authors show the growth curve for each tumor?
The methodology should be revised because most methods lack important information.
In the case of determining release using the HPLC method, there is no information about which HPLC equipment was used. What phase was employed, and exemplary histograms should be presented in the supplementary materials. Has the method been validated?
Could the authors explain the statement "free Cst group released over 90% of Cst within 12 h, which suggested a rapid drug release behavior. However, Cst@TPP-M only released about 18% of Cst, while the release amount of Cst@HA/TPP-M within 12 h was approximately 30%." What does it mean to free the Cst group in the release study?
Figure 4A appears as if the images taken at 4 h were at a different magnification compared to the other time points; furthermore, the image quality is low, and there is a significant background.
The plots of flow cytometry results are incorrectly labeled, specifically regarding FITC-H on the Y-axis. The authors should measure MFI and present the results accordingly.
The results from in vivo imaging should also be presented as quantitative data.
Some sentences should be corrected, for example:
"The cells were evenly spread in the 6-well plate with a density of 2×105 cells/pore". What does it mean "pore"?
Reviewer 4 Report
Comments and Suggestions for Authors
Please, see attached file. One more note: Introduction is too small. It is necessary to widen it by brief description of TNBC features and current trends to combat it. In addition, why in the Methods there is no description of the NMR and IR measurements? Please, add the details about how the samples were prepared and what spectrometer models were used for running the spectra.

Comments on the Quality of English LanguageSee attached file.